# PKIB, a Novel Target for Cancer Therapy

**DOI:** 10.3390/ijms25094664

**Published:** 2024-04-25

**Authors:** Anna Musket, Jonathan P. Moorman, Jinyu Zhang, Yong Jiang

**Affiliations:** 1Department of Internal Medicine, Quillen College of Medicine, East Tennessee State University, Johnson City, TN 37614, USA; musket@etsu.edu (A.M.); moorman@etsu.edu (J.P.M.); 2Center of Excellence in Inflammation, Infectious Disease and Immunity, Quillen College of Medicine, East Tennessee State University, Johnson City, TN 37614, USA; 3Hepatitis (HCV/HBV/HIV) Program, James H. Quillen VA Medical Center, Department of Veterans Affairs, Johnson City, TN 37614, USA

**Keywords:** PKA, cancer, PKIB, cyclic AMP and AKT

## Abstract

The serine-threonine kinase protein kinase A (PKA) is a cyclic AMP (cAMP)-dependent intracellular protein with multiple roles in cellular biology including metabolic and transcription regulation functions. The cAMP-dependent protein kinase inhibitor β (PKIB) is one of three known endogenous protein kinase inhibitors of PKA. The role of PKIB is not yet fully understood. Hormonal signaling is correlated with increased PKIB expression through genetic regulation, and increasing PKIB expression is associated with decreased cancer patient prognosis. Additionally, PKIB impacts cancer cell behavior through two mechanisms; the first is the nuclear modulation of transcriptional activation and the second is the regulation of oncogenic AKT signaling. The limited research into PKIB indicates the oncogenic potential of PKIB in various cancers. However, some studies suggest a role of PKIB in non-cancerous disease states. This review aims to summarize the current literature and background of PKIB regarding cancer and related issues. In particular, we will focus on cancer development and therapeutic possibilities, which are of paramount interest in PKIB oncology research.

## 1. Introduction

Cyclic AMP-dependent protein kinase A (PKA) is a family of holoenzymes that are cyclic AMP (cAMP) effector proteins heavily involved in metabolic processes, including sugar and lipid metabolism, as well as genetic regulation through cAMP response element-binding protein (CREB) [1]. PKA is normally an inactive, tetrameric complex composed of two regulatory (PKA-r) and two catalytic subunits (PKA-c). There are four isoforms of the regulatory subunits R1α, R1β, R2α, and R2β; additionally, there are three isoforms of the catalytic subunits Cα, Cβ, and Cγ [1]. These subunits of PKA-r and PKA-c are differentially expressed in different cell types, leading to cell-specific PKA holoenzyme formation [2]. Following activation by cAMP, the catalytic subunits phosphorylate serine and threonine in other target proteins [3]. Due to the cAMP-mediated nature of PKA activation, PKA is seen as a classical example of an effector protein for cAMP signaling [4]. Additionally, PKA is a highly conserved protein, and both cAMP and PKA are expressed ubiquitously throughout mammalian tissues and cells [5,6]. Cytosolic cAMP acts as a secondary intracellular signaling molecule to bind to the regulatory subunits to activate PKA. Each regulatory subunit can bind to two individual cAMP molecules, for a total of four cAMP molecules binding to activate the PKA heterotetramer. The binding of cAMP to the PKA tetramer results in the activation and disassociation of the PKA-c from the tetrameric complex. It was previously thought that the cAMP-mediated disassociation of the PKA-c from PKA resulted in the activation of the kinase activity [7]. However, it was recently discovered that there are activated tetrameric PKA holoenzymes, indicating that the PKA-c are active while still part of the tetramer complex. More work is needed to determine whether the PKA activation by cAMP initially induces the disassociation of the PKA-c, which can then reform into the tetramer complex, or whether cAMP activates the PKA-c before the disassociation so that the tetramer itself can have kinase activities. Regardless of the timing of the activation compared to the disassociation, the cAMP binding to the PKA-r subunits releases the active PKA-c subunits, allowing them to phosphorylate specific substrate proteins [8].

Once the catalytic subunits have been activated and disassociated from the PKA tetramer, they can participate in several oncogenic processes [9]. One oncogenic pathway regulated by cAMP/PKA is phosphatidylinositol-3 kinase (PI3K) signaling through AKT. In the PKA/AKT pathway, PKA-c phosphorylates AKT at Ser473, which contributes to cancer cell survival. A second oncogenic pathway regulated by cAMP/PKA is the activation of the nuclear transcription factor CREB. The PKA/CREB pathway requires the translocation of cytosolic PKA-c to the nucleus, which first requires the disassociation of PKA-c from the PKA holoenzyme [7]. Thus, it is cAMP-mediated PKA-activation-dependent. The nuclear-localized PKA-c can then bind to CREB, which is bound to the cAMP response element (CRE). CREB can then activate the transcription of various genes responsible for increasing the malignancy of cancers [10]. In addition to the oncogenic PKA/AKT and PKA/CREB signaling pathways, there are also several other impactful signaling pathways regulated by cAMP/PKA [11]. These include regulation of the cell cycle as well as inducing mitogenesis in endocrine cells, both of which are known events in cancer development [12,13,14]. Further, PKA activation augments the Warburg effect, through which cancer cells switch metabolism from mitochondrial oxidation to glucose-dependent glycolysis regardless of the aerobic nature of the microenvironment of the cancer cells [15]. 

The overexpression of PKA is a common characteristic in various types of cancer. Part of the PKA that is formed in cancer cells is secreted and is found as extracellular PKA in the serum of cancer patients. Moreover, recent evidence points to PKA as a viable tool for tumor diagnosis and suggests PKA as a potential target for tumor therapy [4]. For these reasons, inhibitors of the PKA signaling axis are highly desirable, particularly in a cancerous environment or as tumor therapy. Particularly, the use of PKA-specific inhibitors would be of great value to further understand the precise mechanisms of PKA downstream signaling as well as to hinder tumor development and growth. However, due to the systemic abundance of PKA, it is challenging to inhibit only the PKA with carcinogenic functions without adversely affecting the rest of the patients’ systems. Antibody and small-molecule inhibitors that target PKA only work under prohibitively selective conditions, and, often, resistance develops via the expression of a different isoform of the PKA tetramer to negate the effects of PKA inhibition [16,17]. 

Endogenous, small peptide inhibitors (PKI) of PKA were discovered to inhibit PKA-c kinase activities and therefore have important roles in carcinogenesis [18]. There are three isoforms of PKI: PKIα, PKIβ (PKIB), and PKIγ. The most widely characterized concerning cancer treatment and development is PKIα. However, less is known about the impact of PKIB in cancer development and therapeutic strategies. The study of the mechanisms and activities of PKI against PKA in cancer is a growing field, and this review aims to summarize the research specifically about PKIB. Despite the importance of the PKIB inhibition of the PKA pathway, we were unable to find any review papers summarizing the current field of PKIB research. For this reason, we have pulled together some of the most interesting work focusing on PKIB research in cancer biology as well as some tangential PKIB research that may be impactful in cancer. The goal of this review is to highlight the potential of PKIB in cancer research with a focus on elucidating the role PKIB plays in cancer development and possible treatment options.

## 2. The Regulation of PKA Activity by PKI

The three isoforms of PKI are small, heat-stable proteins sometimes called PKA inhibitory peptides due to the short sequences that range from 70 to 75 amino acids long [19]. PKIα and PKIγ are both 75 amino acids in length, whereas PKIB is only 70 amino acids long [20,21,22]. The three PKIs share similar peptide sequences, with the highest similarity being between PKIα and PKIγ, with 96% similarity, whereas PKIα and PKIB share only 40% structural similarity [23,24]. Together, the three isoforms of PKI are known to have nuclear as well as cytosolic functions in oncogenesis and metastasis. All three isoforms of PKI not only have a shared nuclear export sequence but also an affinity to the PKA-c subunits [25]. Thus, the three isoforms of PKI can mitigate nuclear PKA/CREB malignant transformations as well as cytosolic PKA/AKT oncogenic signaling. 

PKIα is the most well-studied PKI and has the highest affinity to PKA-c of the three PKI isoforms [21]. Relatively little is known about PKIγ, but that may be attributed to its high structural similarity to PKIα [20]. PKIB is not as well categorized as PKIα, with little research into the specific effects of PKIB expression. Additionally, while PKIα and PKIγ are nearly as ubiquitous as PKA itself, PKIB has more cell- and tissue-specific expression levels [24,26]. The three PKI isoforms all share some specificity to bind to the PKA-c, but PKIB has the least inhibitory binding effect on PKA-c, thus the focus on PKIα in cancer studies [21]. This may be attributed to tertiary structure differences between the two PKI isoforms, as the inhibitory effect of PKIα has been isolated to a five-amino-acid-long sequence that is shared with PKIB [21]. However, while the tertiary structure of PKIα is known, the tertiary structure of PKIB remains to be determined [27]. 

## 3. PKIB Impacts on Cancer

In this section, the relationships between various cancer types and PKIB will be covered. PKIB has been implicated in several different cancer types as a potential target for cancer therapy, including prostate, breast, lung, and colorectal cancers, which account for half of new cancer diagnoses and nearly half the number of cancer deaths each year [28]. As expected from the common nature of PKA, PKIB is involved in multiple aspects of cancer (Figure 1). In tumor development, the levels of several transcriptional activators have been correlated to PKIB (Figure 1A). These include the nuclear import of PKA-c to interact with CREB on the DNA, as well as linking estrogen hormonal signaling to the activator protein 1 (AP1) on the promotor of the *PKIB* gene (Figure 1A). AP1 is involved in migration, proliferation, and invasion of cancer cells [29]. Additionally, the AP1 transcription complex is heavily involved in T-cell activation, leading to T-cell exhaustion in cancer [30], thus tying PKIB into the well-known carcinogenic potential of the AP1 mechanisms. PKIB is itself implicated in the tumor immune environment through the work of Gerhard et al. [31]. Gerhard’s work showed that *PKIB* is one of the top ten enriched genes in classical dendritic cells [31]. Furthermore, using classical dendritic cells isolated from within tumors of lung, breast, ovaries, and colorectal cancers, Gerhard demonstrated the potential of these dendritic cells to activate both helper T-cells and cytotoxic T-cells in the tumor microenvironment [31]. This study highlights the potential of PKIB as both a diagnostic and prognostic marker in early cancers. 

In addition, genetic regulation in cancer has been linked to PKIB for hypoxia-inducible factors (HIF1α and HIF2α) (Figure 1B). Hormonal signaling outside of the usual G-protein coupled receptor signaling for the generation of cAMP and activation of the PKA pathway has also been linked to PKIB through estrogen and androgen (Figure 1C). Cytosolic interactions have been observed between PKIB and the cAMP/PKA and the PI3K/AKT signaling pathways (Figure 1D). The expression levels of E-cadherin and vimentin, two documented metastatic factors, are regulated by PKIB, thus demonstrating an impact of PKIB expression on cancer metastasis (Figure 1E). Together, the interactions between PKIB signaling and genetic regulation can contribute not only to the development of treatment resistance but also to tumor recurrence in cases where the tumors were successfully treated (Figure 1F,G). 

The impact of PKIB expression varies between cancer types as well as the tissue of origination of the tumor. Cancer genomic and proteomic analyses of PKIB, using The Cancer Genome Atlas program, have shown that the most common PKIB alterations are deep deletions and amplifications [33]. However, there are also rare point mutations and fusion proteins that have been identified, but these are infrequent enough that they have not been correlated with a particular disease state yet. In colorectal cancer, PKIB acts as a tumor suppressor, where the loss of PKIB expression is correlated with a decrease in normal mucosa [34]. In osteosarcoma, PKIB expression is associated with tumor growth but negatively correlated with metastasis [35]. Alternatively, in lung, prostate, and breast cancers, PKIB expression indicates a highly proliferative cancer with poorer patient outcomes including increased mortality [23,36,37]. In lung cancer, *PKIB* was identified as one of the genes correlated with tumor recurrence following surgical resection, although the mechanism for this has not been clarified [32].

## 4. PKIB in Cytosolic Signaling

Several links connect gonadotropic steroids to PKIB expression, typically in the instance of sex hormone-driven oncogenesis. Some of the other cytosolic pathways that are impacted by PKIB are the cellular surface expression of metastatic factors such as E-cadherin and vimentin. Additionally, PKIB contributes to a significant signaling cascade for the field of cancer biology, the well-known PI3K/AKT (AKT is sometimes called PKB or Rac) signaling cascade. Notably, for carcinogenesis, PI3K/AKT signaling is activated by growth factor and growth factor receptor signaling and contributes to the evasion of apoptosis and proliferation of cancer cells. G-protein coupled receptors (GPCR) have also been linked to PKIB in the cytosol.

In osteosarcoma, the expression of E-cadherin and vimentin was changed by induced overexpression of PKIB [35]. This is the first study to date that examines the relationship between metastasis and PKIB protein expression. Wan et al. found that increased expression of PKIB was correlated with more proliferative tumor growth but also a decrease in metastatic potential. Overexpressing PKIB in osteosarcoma cells downregulated E-cadherin but upregulated vimentin [35]. Further, PKIB overexpression significantly decreased the ability of the osteosarcoma cells to migrate and invade adjacent tissues. This study also contributed to the literature demonstrating the link between PKIB expression and the phosphorylation of AKT. 

PKIB is known to regulate the PKA-c phosphorylation of AKT at Ser473 (AKT^Ser473^); however, the exact mechanism for this phosphorylation is unknown [23]. It is yet unclear if the overexpression of PKIB causes increased phosphorylation of AKT^Ser473^ or if increased phosphorylation of AKT^Ser473^ causes an increase in PKIB expression. The Dabanaka et al. study showed a strong correlation between PKIB, phosphorylated AKT^Ser473^, as well as triple-negative breast cancer morphology, but the mechanisms correlating PKIB and AKT^ser473^ were not investigated. However, there is some evidence that suggests that PKIB phosphorylates AKT^Ser473^ through a joint effort with PKA-c [38]. What is clear is that higher expression of PKIB correlates with increased phosphorylation of AKT^Ser473^ across several different cancer types, including osteosarcoma, breast, lung, and prostate cancers [23,35,36,37,39]. Additionally, the inhibition of PI3K reversed this PKIB-induced phosphorylation of AKT^Ser473^ in non-small-cell lung carcinoma (Figure 2E) [37]. Further, the silencing of PKIB was achieved through the induction of microRNA495, which attenuated the PKIB-induced phosphorylation of AKT^Ser473^ in a murine model of irritable bowel syndrome [38]. Together, these results imply a correlation between PKIB expression and activation of the highly oncogenic signaling of the PI3K/AKT pathway that requires further study to be conclusive. It would be interesting to see the effect microRNA495 has on other models of PKIB, such as Gerhard’s study highlighting *PKIB* as a potential biomarker for early breast, lung, ovarian, and colorectal cancers [31]. Additionally, the potential impact of microRNA495 on the PKIB/AKT axis would be of great value in cancer research and treatment.

Through the modification and regulation of these three signaling pathways, estrogen/androgen, E-cadherin/vimentin, and the PI3K/AKT cascade, PKIB has important cytosolic roles in cancer development, metastasis, and proliferation/evasion of apoptosis, respectively (Figure 2D). PKIB expression is often correlated with decreased cancer patient prognosis, with one exception. In normal colorectal mucosa, PKIB is highly expressed; however, during the transition from normal mucosa to colorectal cancer, PKIB expression is lost [34]. This indicates a protective function of PKIB that is suppressing tumors, but the mechanisms for this protective effect are unclear. Yet, these results call into question the potential of *PKIB* as a potential biomarker for colorectal cancer, as shown by Gerhard et al. [31]. Perhaps this can be explained by the addendum in Gerhard’s work that the ten potential biomarkers were all identified in the early stages of the cancers and therefore *PKIB* may not be indicative of cancer in the colorectal instance. Further work is needed to clarify this discrepancy. 

GPCRs are known to interact with the PKA pathway through the simple fact that the GPCR signaling pathways produce cAMP and therefore activate PKA signaling [40]. PKIB colocalizes to a GPCR, GPR39, to increase the GPCR activity [41]. This PKIB/GPR39 interaction on the inside of the cell membrane increased the GPCR signaling in a ligand-independent manner and protected from oxidative stress [41]. The addition of zinc to cells with the GPR39/PKIB colocalization freed PKIB from the GPR39 and enabled PKIB to interact with PKA-c, forming a negative feedback loop that can limit the GPCR signaling pathways [41]. Interestingly, when the PKIB domain that binds to AKT and PKA-c was removed from a cloned version of PKIB, the colocalization of PKIB and GPR39 still occurred, whereas the PKA-c interactions did not, demonstrating that there is a second PKIB domain capable of binding to other proteins that remains to be determined [41].

## 5. PKI in Cancer

Despite the multiple cytosolic properties and interactions of PKIB, the impact of PKIB expression is not solely a cytosolic phenomenon but also has a cellular nuclear effect. Three genetic regulators of PKIB expression have been identified in various cancers. The first is hypoxia-inducible factors one and two α (HIF1α and HIF2α). HIF1α is known to slightly upregulate PKIB, and HIF2α strongly upregulates (greater than 2-fold increase) both PKIB and PKIα [42]. Interestingly, while HIF1α does upregulate PKIB, it does not upregulate PKIα, despite the strong similarity between the *PKIα* and *PKIB* genes [42]. Further research into the link between PKIB and HIF1α demonstrated that the *PKIB* promoter is bound to HIF1α in beta cells so that increasing amounts of HIF1α show a corresponding increase in the expression of PKIB (Figure 2C) [43].

In addition to the hypoxia-inducible factors, there are two other known regulators of *PKIB* transcription and expression. The nuclear estrogen receptor alpha (ERα) and cFos, part of the AP1 transcription factor complex, bind to the same region of the *PKIB* gene and increase protein expression in breast cancer (Figure 2A) [44]. The treatment of Erα-positive breast cancer cells with 17β-estradiol (an estrogen steroidal hormone) increased the expression of PKIB. An siRNA knockdown of PKIB in this study resulted in a decrease in the proliferation of breast cancer cells, thus confirming the role of nuclear estrogen signaling and PKIB in the proliferative potential of breast cancer. In a separate study, Yang et al. suppressed the expression of PKIB as a target gene of ERα signaling in breast cancer cells [45]. In this work, the ERα signaling could be controlled and stabilized by a deubiquitinase enzyme, and the downstream expression of PKIB was also suppressed [45]. A study by Creevey et al. further confirmed the role of gonadal hormone signaling on the *PKIB* gene promoter by using androstenedione, a weak androgen/estrogen precursor steroid, to treat breast cancer cells [46]. Following treatment, the breast cancer cells were analyzed to look for transcriptomic changes related to the androgen and estrogen receptors. *PKIB* was one of the genes that was regulated by the binding of androgen receptors to androstenedione [47]. Together, the estrogen receptors and androgen receptors are both linked to an increase in PKIB expression, with a correlating increase in proliferation and treatment resistance developing in the breast cancer cells, promoting the EGFR signaling pathway and autophagy [47].

The AP1 complex component cFos also binds to the *PKIB* gene and increases PKIB expression in breast cancer [44]. In addition to the direct cFos binding to *PKIB,* the cAMP response element (CRE) is adjacent to the leucine zipper component of the AP1 transcription factor complex [30]. Together, both the direct binding of cFos/*PKIB* and the potential for nuclear PKA-c/PKIB complexes to occur imply that PKA-c/PKIB can be directly shuttled to the cFos/*PKIB* and form a positive feedback loop, leading to increasing upregulation of PKIB. Moreover, the adjacence of CRE to the leucine zipper component of AP1 further implicates a potential role of PKIB-mediated feedback leading to increasing cancer progression. However, while the individual links (cFOS/*PKIB,* CRE/PKIB, AP1/PKIB) have been demonstrated, the mechanistic links between AP1, cFOS, CRE, and PKIB have not been confirmed to co-occur. More work is needed on the nuclear functions of the *PKIB* gene and PKIB protein in the CRE/AP1 complex in the cancerous state.

## 6. PKIB Research with Indirect Cancer Impact

Most studies with PKIB have focused on cancer research, with an emphasis on cancers that are sex hormone-driven, such as prostate and breast cancers. However, there has also been some insight into the role of PKIB in non-cancerous environments, such as embryonic development, metabolism, immune and checkpoint regulation, diabetes development, and even irritable bowel syndrome [38,43,48,49,50,51]. All of these topics are influenced in some manner by the presence, or conspicuous absence, of PKIB. Together, the non-cancerous impacts of PKIB appear to continue the trend of either deep deletions or overexpression of PKIB.

In embryonic models of Huntington’s disease, *PKIB* was shown to maintain steady levels throughout neuronal developments in wildtype stem cells, but in stem cell models of Huntington’s disease, *PKIB* expression fluctuated throughout the stages of differentiation [50]. Following the stem cell model of cancer development, it would be interesting to see how the levels of PKIB change, if at all. In β-cells, chronic hyperglycemia induces PKIB expression, which further leads to insulin resistance [43]. PKIB interacts with the immune system in several ways, from the PKIB-expressing classical dendritic cell-induced activation of T-cells to the AP1/PKIB-induced immune system interactions, which could both have far-reaching consequences in cancer [30,31]. Hyperglycemia promotes PKIB expression, which leads to increased insulin resistance and could have an impact on cancer metabolism [43]. Inflammatory bowel disease has PKIB overexpression, and as cancer is associated with inflammation, there may be some crossover [38]. There is simply not enough known about the exact mechanisms of a variety of disease states with altered PKIB to draw any conclusive ideas. 

Several of the non-cancerous impacts of PKIB are made more interesting due to their relevance to carcinogenesis and maintenance of the cancerous state. In particular, embryonic development, metabolism, and immune and checkpoint regulation are all key research foci in cancer development. The link between embryonic development and cancer is quite clear, as cancer can be simply defined as a resurgence of the stem cell nature of cancerous cells. Metabolism, of course, plays a key role in the energy requirements, growth potential, and Warburg effect in solid-tumor-forming cancers. Immune and checkpoint regulation is particularly relevant to cancer as well as cancer treatment resistance. The tumor microenvironment is heavily influenced by immune cells and immune cell penetration, so PKIB interactions in both tumor and immune cells may be synergistic.

## 7. Discussion

In prostate cancer, resistance to androgen deprivation treatment increases the amount of 17β-estradiol synthesis. As illustrated in Figure 2A, the increase in 17β-estradiol results in an increase in PKIB expression through the ERα nuclear signaling and transcription activation [44]. Resistance to androgen deprivation treatment is one of the key factors in the progression of prostate cancer to the more lethal castration-resistant prostate cancer. Castration-resistant prostate cancer usually becomes metastatic, although it is not yet known why this metastatic development transition occurs. Further, while the exact mechanisms between ERα signaling and the development of castration-resistant prostate cancer are unknown, PKIB may be a part of that missing mechanism between ERα and the development of castration-resistant prostate cancer, whether metastatic or not. The link between estrogen signaling, PKIB, and castration-resistant prostate cancer is further indicated by the study by Chung et al. Chung’s study demonstrated that the induced overexpression of PKIB in prostate cancer increased the aggressiveness of the cancer cells, whereas the knockdown of PKIB decreased the cancer cells’ aggressiveness [36]. 

In osteosarcoma, PKIB was linked to the expression of two well-known epithelial–mesenchymal transition proteins, E-cadherin and vimentin [35]. However, it would be negligent to think that the results shown in osteosarcoma cells would be replicable in all cancer types. It would be particularly interesting to see how primary tumor sites and secondary metastatic locations differ in PKIB expression, especially considering the transition from mesenchymal back to epithelial cancer cells in a distant location. Also noteworthy is that while the majority of research has thus far shown a detrimental effect of PKIB, the opposite effect is seen in colorectal cancer [34]. PKIB is expressed only in normal colorectal cells, and the expression of PKIB is lost during the transition from normal tissues to cancerous tissues. In this instance, PKIB appears to act as a tumor suppressor, although no mechanism for this activity has been shown yet. This research by Lacalamita et al. [34], in particular, demonstrates the complexity of PKIB interactions and highlights that PKIB expression and effects are likely to be not only organ-specific but also disease-specific. However, as established by the non-cancerous impacts of PKIB in various tissues, the influence of PKIB may also vary between cell types. 

## 8. Conclusions

While there is scattered knowledge about the multi-variate roles of PKIB in cancer and cancer development, there are few firm conclusions that can be drawn to paint PKIB as either oncogenic or protective. However, more evidence exists that points towards the oncogenic functions of PKIB in a variety of cancer types. Furthermore, while there are no known pharmaceutical inhibitors of PKIB, the microRNA495 shows potential for studying PKIB interactions. With a PKIB-specific inhibitor, it would be easier to understand the interactions and mechanisms of PKIB-altered diseases and cancers. The lack of experimentally derived protein folding structure further complicates the continuation of PKIB research. Without a tertiary structure of PKIB, it becomes more difficult to determine mechanistic interactions between PKIB and effector proteins. Thus, PKIB requires further investigation into both the healthy and cancerous states to fully understand the relevance in carcinogenesis and cancer treatment. There is accumulating evidence that suggests that endocrine-driven cancers, such as Erα-positive breast cancers, may have a greater sensitivity to PKIB alterations than do other cancer types. However, there is still much to be learned from PKIB interactions that relate to many different signaling pathways and genetic functions. For example, the potential of PKIB as a cancerous biomarker provides another possible target for multi-cancer treatment. Together, there are many possibilities for PKIB research and applications, giving PKIB a hopeful future in cancer biology. 

## Figures and Tables

**Figure 1 ijms-25-04664-f001:**
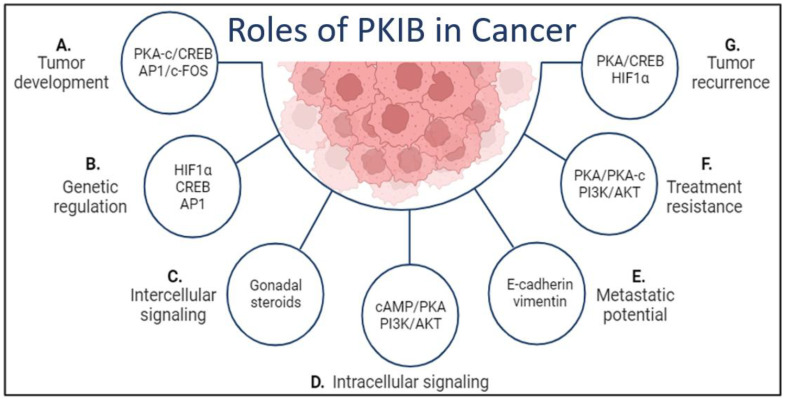
The roles of PKIB in cancer development and tumor microenvironment. (**A**). Two transcription activation protein complexes can be modulated by PKIB, the PKA-c/CREB complex and the AP1/c-FOS complex. (**B**). Three genetic regulators are known to interact with PKIB: HIF1α, CREB, and AP1. (**C**). PKIB signaling can be mediated by intercellular components that are in the tumor microenvironment, particularly in estrogen-/androgen-driven cancers. (**D**). Cytosolic signaling pathways that interact with or are mediated by PKIB include the key cAMP/PKA and the PI3K/AKT pathways. (**E**). PKIB may have the potential to mediate the epithelial–mesenchymal transition and effect metastasis through the regulation of E-cadherin and vimentin. (**F**). PKIB interacts with and modulates the signaling for both the PKA and AKT signaling cascades. (**G**). Although the exact mechanism between PKIB expression and tumor recurrence has not been studied, it is likely to be due to PKIB interactions with a variety of signaling pathways, both nuclear and cytosolic [32]. Figure created in BioRender.

**Figure 2 ijms-25-04664-f002:**
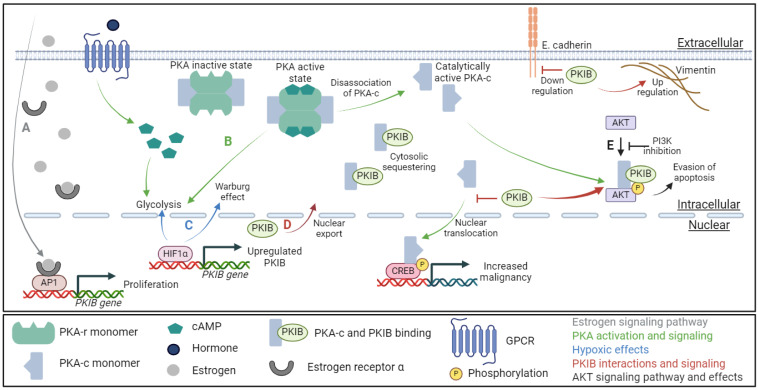
The PKIB oncogenic signaling pathway interactions. (**A**). In the estrogen receptor pathway (gray arrow), cytosolic estrogen bound to ERα translocates to the nucleus, where it can interact with AP1 on the promotor of PKIB to increase the expression and proliferation potential of PKIB. (**B**). In the PKA pathway (green arrows), GPCR signaling activates PKA through cAMP; the PKA-c subunit then disassociates from the PKA complex and activates AKT in the cytosol and CREB in the nucleus. (**C**). Under hypoxic conditions (blue arrows), HIF1α upregulates PKIB protein expression as well as increases the Warburg effect by modulating glycolysis. (**D**). Representative PKIB signaling (red arrows/inhibitors) showing nuclear PKIB being exported to the cytosol where it interacts with PKA-c to modulate AKT phosphorylation while maintaining PKA-c in the cytosol to inhibit PKA-c/CREB transcriptional activation. It is also demonstrated that PKIB decreases E-cadherin while increasing vimentin to reduce the metastatic potential in osteosarcoma. (**E**). Oncogenic AKT signaling pathway (black arrows/inhibitors) showing that the inhibition of PI3K mitigates the phosphorylation of AKT by PKA-c/PKIB. Figure created in BioRender.

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
