# Peer review of "PKIB, a Novel Target for Cancer Therapy"

_ijms, 2024, doi:10.3390/ijms25094664_

Round 1

Reviewer 1 Report

Comments and Suggestions for Authors

Authors proposed a paper entitled “PKIB, a novel target for cancer therapy”. The paper has been proposed for the IJMS.

The paper has a good scientific soundness, but it requires some additions and modifications before publication to this Journal.

Authors are encouraged to add missing elements to the abbreviation list, due to the large number of acronyms used in this work.

Reguarding the content expressed in the first introductive paragraphs, the Introduction needs to be improved with a larger number of references.

About the final part of the introduction, there should be a clear definition of the goals and main aims of this review paper.

Line 96. Please check this line since “(Thomas et al., 1991).” Is not reported as [x]. Please check also the reference style, according to the guidelines of this Journal.

Figure 1. I would try to improve the focus of this figure, especially reguarding the written parts. Maybe authors could consider to reduce the written parts in the figure and better describe them in the main text or in the figure caption. For example, in the figure area, I would only indicate the letters, and describe the letters association in the caption of the figure.

I believe that the title of paragraph “6” as “Non-cancer Research into PKIB” is not correctly giving the intended idea. I suggest substituting it.

Lines from 295 to 300 could be separated and named as “conclusion” section, that is missing in the document. However, some final remarks should be added to this newly introduced ection, together with an expanded concepts about future perspectives.

Comments on the Quality of English Language

A quite good use of English.

Reviewer 2 Report

Comments and Suggestions for Authors

ijms-2953935, PKIB, a novel target for cancer therapy, by Anna Musker et all

The manuscript is a short review of the literature on PKIB as anticancer target. The work could be useful for the journal’s readers, but there is a lot of room for improvements.

The number of references is very small in my view and I think it could be higher if the authors would expend their work beyond the general focus of the paper. The authors should present a section on drugs that are targeted towards PKIB. If there are similar review works on PKIB, please highlight in the introduction section what does this paper adds new and its importance in the field.

The manuscript lacks a critical approach. It looks more like a collection of data, and less than a proper review. The authors should try to present an analysis of the potential use of this target in therapy. Are there advantages? Opportunities? Challenges? What are the main types of cancer that could benefit? But those that would be resistant? What would be the main problems associated with the pharmacological use of PKIB and how would be resolved? Are there different opinions in the field?

The authors try in some section to discuss different possibilities, like “It is yet unclear” or “there is some evidence that suggests”. I think these sections add value to the paper and they should expend them. Discuss further if the evidences are convincing or if there might be some other possibilities or if the referred study is biased or problematic.

It would be interesting to view the authors opinion based on their expertise on the field.

Comments on the Quality of English Language

OK

Reviewer 3 Report

Comments and Suggestions for Authors

Although the topic chosen for the review is very interesting, the bibliographic review on PKIB has remained vague and very limited. PKIB has innumerable functions both in cancer and in many other biological events that the authors list very briefly. If the authors want to focus on the functions of PKIB related to cancer, they must delve deeper into the mechanisms and their possible repercussions on treatments, etc., and specify the different types of cancer. If the authors want to provide information about what we know in other biological aspects, they should delve deeper into them and change the title of the review.

Reviewer 4 Report

Comments and Suggestions for Authors

This is an interesting review article with adequate novelty amd quality. Some points should be addressed.

- In the Abstract, line 18, the authors should report some example of transcription factors ot other proteins that were regulated by PKIB.

- In the Introduction section, the authors should split the first paragraph into two separate paragraphs (maybe in line 38?).

- In the Introduction section, the authors could add the word "also" in line 66. e.g.,  ".... pathways, there are also several other .....".

- In line 177, the reference should be numbered.

- The sentence in line 182-186 could be spit into two sentences in order to be more clearly inderstood for the readers.

- The sentence in lines 255-258 is quite complex and confusing and needs re-writting.

- The sentence in lines 275-280 could be split into two sentences in order to be more clearly understood for the readers.

- A conclusion section is missing. This section is very important for the readers.

Overall, this review article is very well-written and very well-organized.

Comments on the Quality of English Language

Minor editing of English language is recommended.

Round 2

Reviewer 1 Report

Comments and Suggestions for Authors

Authors provided a revised version of their paper. According to my opinion, it can be accepted for publication.

Author Response

Thank you for reviewer's comments.

Reviewer 2 Report

Comments and Suggestions for Authors

The revised work is improved and it has the necessary quality for it to be published.

Comments on the Quality of English Language

OK

Author Response

Thank you for reviewer's comments. 

Reviewer 3 Report

Comments and Suggestions for Authors

Sorry, but the authors could remark the changes in the new manuscript please? And the figures should be reviewed.

Author Response

Thank you for reviewer's comments. We have remarked the changes in the new manuscript with yellow highlighted color. We also reviewed the figures in our manuscript.